# Genome Wide Association Study of Karnal Bunt Resistance in a Wheat Germplasm Collection from Afghanistan

**DOI:** 10.3390/ijms20133124

**Published:** 2019-06-26

**Authors:** Vikas Gupta, Xinyao He, Naresh Kumar, Guillermo Fuentes-Davila, Rajiv K. Sharma, Susanne Dreisigacker, Philomin Juliana, Najibeh Ataei, Pawan K. Singh

**Affiliations:** 1ICAR- Indian Institute of Wheat and Barley Research, Karnal 132001, India; 2International Maize and Wheat Improvement Center (CIMMYT), Carretera México-Veracruz Km. 45, El Batán, Texcoco, CP 56237, Mexico; 3ICAR- Indian Agricultural Research Institute, New Delhi 110012, India; 4INIFAP-CIRNO, Campo Experimental Norman E. Borlaug, Apdo. Postal 155, Km 12 Norman E. Borlaug, Cd. Obregon, Sonora, CP 85000, Mexico; 5Agricultural Research Institute of Afghanistan (ARIA), Kabul 1001, Afghanistan

**Keywords:** Karnal bunt, *Neovossia indica*, Resistance breeding, SNPs, *Triticum aestivum*

## Abstract

Karnal bunt disease of wheat, caused by the fungus *Neovossia indica*, is one of the most important challenges to the grain industry as it affects the grain quality and also restricts the international movement of infected grain. It is a seed-, soil- and airborne disease with limited effect of chemical control. Currently, this disease is contained through the deployment of host resistance but further improvement is limited as only a few genotypes have been found to carry partial resistance. To identify genomic regions responsible for resistance in a set of 339 wheat accessions, genome-wide association study (GWAS) was undertaken using the DArTSeq^®^ technology, in which 18 genomic regions for Karnal bunt resistance were identified, explaining 5–20% of the phenotypic variation. The identified quantitative trait loci (QTL) on chromosome 2BL showed consistently significant effects across all four experiments, whereas another QTL on 5BL was significant in three experiments. Additional QTLs were mapped on chromosomes 1DL, 2DL, 4AL, 5AS, 6BL, 6BS, 7BS and 7DL that have not been mapped previously, and on chromosomes 4B, 5AL, 5BL and 6BS, which have been reported in previous studies. Germplasm with less than 1% Karnal bunt infection have been identified and can be used for resistance breeding. The SNP markers linked to the genomic regions conferring resistance to Karnal bunt could be used to improve Karnal bunt resistance through marker-assisted selection.

## 1. Introduction

Wheat (*Triticum aestivum* L.) is one of the most important staple cereals grown worldwide contributing about 21% of the energy intake of the world population [1]. Karnal bunt disease of wheat caused by the fungus *Neovossia indica* (Mitra) Mundkur (syn. *Tilletia indica*), is an important disease related to international trade. The disease is characterized by the replacement of a part of the seed with a black powdery mass of spores. This disease affects not only the grain weight but also the grain quality of wheat due to the production of trimethylamine with an unpleasant rotten fishy smell [2]. Karnal bunt was first reported in 1931 from Karnal town in the Indian state of Haryana [3]. Since then, the disease has spread to different states of India, as well as Nepal [4], Iran [5], Iraq [6], Afghanistan [7], Mexico [8], limited areas of the USA [9], South Africa [10] and Brazil [11]. Climate change, trade liberalization, globalization and international transportation of people and commodities have increased the potential for disease incursion [12,13]. Once this pathogen is established in the soil, it is very difficult to eradicate [14,15]. Countries such as Australia, USA, EU countries and China have imposed strict quarantine regulations regarding the import/export of wheat grains to prevent the accidental entry of the pathogen [16,17].

The losses in grain yield by the disease are marginal in the areas infested with this disease; however, it greatly reduces the quality and marketability of the product. The estimated losses due to this disease in Sonora and Sinaloa, Mexico for the years 1982–1989 were estimated to be about MN$ 16,689 million [18]. Similarly, in the USA, it was estimated that the cumulative reduction of national farm income from 2003–2007 relative to the baseline was USD 5.3 billion [19]. The loss of export market due to Karnal bunt in any country would lead to reduction in production and price, ultimately reducing the farm income.

The disease has been contained by making use of crop rotations, through the use of disease free seed and adjusting the time of irrigation, as well as by making use of chemical fungicides [14]. The most feasible approach has been to employ host resistance available in the germplasm. Genetics of Karnal bunt resistance indicates that the resistance is governed by one to several genes having partial dominance [20,21,22,23,24,25]. Some of the Indian, Chinese and Brazilian wheat have been reported to have sufficient levels of resistance to Karnal bunt [26,27]. Besides cultivated germplasm, several accessions of *Aegilops* species [28] and synthetic hexaploid wheat (derived from hybridization of *T. durum* and *Ae. taushii*) were found to be Karnal bunt resistant [29]. Six resistance genes, *1 to 6,* have been designated in different breeding lines [21] but the chromosomal location of any of the designated genes is unknown. Molecular investigations have reported QTL conferring resistance to Karnal bunt on different wheat chromosomes (Table 1) [30,31,32,33,34,35,36,37,38].

Only few Karnal bunt resistant sources have been identified and used in the breeding programs, making it necessary to explore new sources of resistance and enrich the breeding pool to broaden the genetic base of the resistance. Most of the known resistance gene(s)/QTL have been identified based on bi-parental mapping populations; thus, the usefulness of the linked markers in materials other than the populations studied remains unknown. Genome wide association study (GWAS) offers unique opportunities to use a diverse set of germplasm that have accumulated a much larger number of crossing over events since their last common progenitor, as opposed to only one or a few meiotic recombinations in a bi-parental mapping populations derived from crosses of two parents only [39]. Good resolution of the identified QTL is often achievable and wide sampling of molecular variation is possible in GWAS. Its ability to resolve marker/trait associations depends upon the extent of linkage disequilibrium (LD) present in the association panel. The genetic relationship among entries in association panel influences LD, which can be easily inferred from genome-wide genotypic datasets [40]. The markers identified to be linked with QTL have the potential to be used across breeding material for identification and introgression [41]. The sequence information, the annotated, whole-genome sequence of wheat and other cereals offer new opportunities to study the exact nature of allelic variation, explore the underlying genetic basis and putative gene(s) associated with Karnal bunt resistance. In wheat, GWAS has been used to study disease resistance like rusts and powdery mildew, drought tolerance and end use quality traits, whereas only few studies were undertaken for Karnal bunt resistance. In this study, a diverse collection of germplasm from Afghanistan including 339 lines has been subjected to GWAS analysis, in order to identify QTL for Karnal bunt resistance.

## 2. Results

### 2.1. Karnal Bunt Resistance in the Germplasm Collection

The percent Karnal bunt infection score in the evaluated germplasm collection ranged from 0% to 60%. In all four experiments, Karnal bunt infection was skewed towards lower infection but continuous variation was observed (Figure 1). Disease severity in the four experiments was highly correlated, with phenotypic correlations ranging from 0.61 to 0.70.

Based on the average of all four experiments, 52 genotypes exhibited a severity less than 5%, 17 genotypes exhibited more than 40% and rest of the lines recorded between 5–40% (Figure 1). It is noteworthy that 11 lines showed severities less than 1% and could be used as Karnal bunt resistance sources (Table 2). Analysis of variance indicated significant variation among genotypes, as well as for genotype-by-experiment interaction (Table 3). The heritability estimate for Karnal bunt was observed to be 0.80 based on all four experiments.

### 2.2. Population Structure

Principal component analysis based on the genotypic data was able to differentiate the germplasm collection into two main groups (Figure 2). Genotypes of International Maize and Wheat Improvement Center (CIMMYT) origin and those derived from CIMMYT lines, as well as some Afghanistan materials, were clustered in Group I, whereas the rest of Afghanistan materials were found in Group II. The lines released in India were also clustered into Group I, in agreement with their CIMMYT based pedigrees.

### 2.3. Markers Significantly Associated with Karnal Bunt

GLM and MLM models were tested for best fit based on the observed *p*-values and the expected *p*-values for a trait (Q–Q plots). The mixed linear model (MLM) involving population structure (Q) and kinship data (K) fitted better than GLM and therefore was used for further association analysis. The association analysis identified a total of 18 marker-trait associations (MTAs) for Karnal bunt. The SNPs exhibiting MTAs were assigned to 12 wheat chromosomes regions, namely, 1D, 2B, 2D, 4A, 4B, 5A, 5B, 6A, 6B, 7B and 7D (Figure 3). Out of the 18 MTAs, only the one on chromosome 2BL was common among all four experiments, the one on chromosome 5BL was common among three experiments, and the remaining 16 were only common between two experiments (Table 4). The identified SNPs explained the phenotypic variation ranging from 5% to 20% for Karnal bunt severity. Four MTAs were reported on chromosome 2B, followed by three on chromosome 5B, two each on chromosome 5A and 6B and only one on chromosomes 1D, 3B, 4A, 4B, 6A, 7B and 7D. The SNP markers showing significant MTAs were tested for linkage disequilibrium (LD) among each other, and the results showed that two SNPs on 5BL (1079540 and 1083023) were in LD, indicating that they may represent the same QTL. The QTL on chromosome 2BL (6048838) explained the highest phenotypic variation from 11 to 20% across the four experiments, and lines having the CC genotype of the marker 6048838 showed an average Karnal bunt infection score of 16.5%, whereas those having TT genotype showed an average infection of 25%. QTL represented by markers 1079540 and 1083023 on 5BL explained a phenotypic variation of 6 to 9% across three experiments, and at 1079540, the CC genotypes showed 17% less Karnal bunt infection as compared to lines having the AA genotype (Figure 4). Similarly, for other significant SNPs, the resistance alleles showed their association with Karnal bunt reduction as well.

### 2.4. Putative Genes Associated with MTAs for Karnal Bunt

In the present study, significant and stable SNPs have been reported on chromosomes 1D, 2B, 3B, 4A, 5A, 5B, 6A, 6B, 7B and 7D. The SNP markers 6048838 and 1228074 on chromosome 2BL encompass an interval of 38Mb and are reported to contain several resistance genes coding for Kinase-like proteins, nucleotide-binding site-leucine rich repeat (NBS-LRR) class of proteins, a serine threonine protein kinase that are mostly associated with disease resistance. The genes TraesCS2B02G496800 and TraesCS2B01G535000, which code for disease resistance protein (Toll-interleukin receptor-NBS-LRR class) and 70 kDa heat shock protein are adjacent to the markers 6048838 and 1228074, respectively. The SNPs on chromosome 5BL (1079540 and 1083023) encompass 14 Kb region and the putative candidate genes in the region were TraesCS5B01G506000 and TraesCS5B01G506100 which code for F-box family protein and F-box domain containing protein, respectively. Another QTL on chromosome 4AL harbors gene TraesCS4A01G379200 coding for F-box family protein. Gene TraesCS1D01G407200 coding for Leucine-rich repeat receptor-like protein kinase family was found in the vicinity of the QTL on 1DL. Similarly, two SNPs (1002872 and 1107259) on chromosome 2BS encompass genes TraesCS2B02G197000 and TraesCS2B01G197500, which code for Kinase family protein and GRF zinc finger family protein.

SNPs on chromosome 6AS and 6BS were found to be close to genes coding for NBS-LRR-like resistance proteins. The remaining SNPs on chromosomes 2D, 5AS, 5AL, 5BS, 6BL, 7BS and 7DL were also found to be associated with Serine/threonine-protein kinase, Protein Kinase family protein, Kinase family protein, Receptor-like kinase, C2H2-like zinc finger protein, Glycosyltransferase and Transcription factor gene families. The candidates genes located nearby/ in proximity to the significant SNP markers are presented in Figure 5. 

## 3. Discussion

Karnal bunt in wheat is of utmost importance with respect to the strict quarantine laws in countries across globe. Host resistance to this disease is of quantitative nature and only few resistance sources are identified and deployed in breeding programs across China, India, Mexico and Brazil [24,31,33,42] (Table 1). The current study made use of a germplasm collection comprising of Afghan landraces and breeding lines mostly of CIMMYT origin disseminated to the country to identify new Karnal bunt resistance sources that can be integrated into the breeding programs to diversify resistance to this disease. The disease development is highly affected by environmental conditions; thereby, it is a challenge to properly screen the genotypes for Karnal bunt resistance [43]. In the present study, correlation and heritability of the %Karnal bunt infection score across experiments was high despite significant genotype-by-experiment interaction, indicating screening and scoring was appropriate for Karnal bunt, which is in agreement with other studies [33,37,38]. Here, we have reported 11 lines exhibiting less than 1% Karnal bunt infection score, and based on their diverse pedigree and different QTL composition, they represent diverse resistance donors for Karnal bunt and thus could be used in resistance breeding (Table 2). Out of these eleven genotypes, one genotype belonged to Afghanistan wheat collection (#6) and the other 10 are of CIMMYT origin based on pedigree information.

Most of the earlier studies for Karnal bunt resistance were based on few lines like ALDAN “S”/IAS 58, H567.71 and W485, for identification of QTL, which were mostly reported on chromosomes 4A, 4BL, 5BL and 6BS [30,31,33,34,37] (Table 1). Results from bi-parental QTL mapping and GWAS are often not corresponded [44,45] but complementary to each other [46,47].

Two genetic loci associated with SNPs identified on chromosomes 2BL and 5BL were the most repeatable, explaining a variation of approximately 14% and 7%, respectively. The 2BL QTL identified in this study is at a distance of 18 Mb from the previously reported *QKb.*cim-2BL [37] and has a TIR-NBS-LRR class protein-coding gene in its vicinity, which is well-recognized for its role in conferring disease resistance [48]. Another QTL on chromosome 5BL was found to be about 37 Mb away from an earlier reported QTL (*Qkb.ksu-5BL.1*) on chromosome 5BL [33,34] and the genes in the vicinity of this QTL code for F-box family protein and F-box domain containing proteins.

The QTL on chromosome 4B was found to be in the same position as the ones identified in previous studies [31,33,34,35]. The loci on chromosome 5AL and 6BS reported in the present study are in close proximity of the previously reported QTL on the respective chromosomes [33,36,37]. The genomic regions on chromosomes 1DL, 2BS, 2DL, 4AL, 5AS, 6BL, 6BS, 7BS and 7DL are found to be potentially new and could be useful for enhancing Karnal bunt resistance in the breeding germplasm. The critical *p* value (0.001) adopted in this study was based on a balance between false positive and false negative. The *p* value would be 3.8 × 10^−6^ if the Bonferroni correction were used, which is too low and would result in the absence of any significant QTL (see Figure 3). As demonstrated in the manuscript, we have identified several QTLs that have been reported previously, which implies that our analysis is robust, and also our critical *p* value is appropriate.

The results of our and earlier studies indicate that high diversity for Karnal bunt resistance exists in wheat germplasm. Although most of the identified MTAs showed low phenotypic effects, a few loci with large affects was still identified, such as the one on 2BL, which could be transferred together with minor QTL into individual backgrounds through MAS for developing Karnal bunt resistance cultivars.

## 4. Materials and Methods

### 4.1. Experimental Plant Material and Field Trials

The plant material consisted of 339 wheat genotypes representing a collection from Afghanistan, which includes landraces, elite lines, released varieties and advanced breeding lines. The experimental material was planted at the Norman E. Borlaug Experimental Station (CENEB) of CIMMYT, located in Cd. Obregon, Sonora (27.4828° N, 109.9304° W). The experiments were conducted for two consecutive crop seasons (2016-17 and 2017-18) by planting genotypes in two row plots of 1 m length on 80 cm raised beds. The Karnal bunt susceptible check WL711 was seeded in a range of dates to monitor disease pressure over the whole experiment period. The field trials were managed as per recommended local practices. The material was planted on two different dates, with the first being in mid-November and the second two weeks later in each year. The heading date and disease development is affected by environmental conditions therefore each planting date was considered as separate experiment (Appendix A). In all there were four experiments, namely, E1-2017 (2017-seeding date 1), E2-2017 (2017-seeding date 2), E3-2018 (2018-seeding date 1) and E4-2018 (2018-seeding date 2) used for screening germplasm collection for percent Karnal bunt infection.

### 4.2. Phenotyping for Karnal Bunt Infection

#### 4.2.1. Inoculum Preparation

To prepare inoculum, Karnal bunt infected wheat kernels were mixed with Tween 20 solution in a glass tube, which was shaken and then filtered with 60 μm mesh and allowed to stand for 24 h. The collected teliospores were placed in 0.6% sodium hypochlorite for 2 min and centrifuged at 3000 rpm for a few seconds. The supernatant was discarded and the teliospores were rinsed with distilled water, then the solution was shortly centrifuged at 3000 rpm. The rinse and centrifuge steps were repeated one more time. The teliospores were transferred to 2% water agar under sterile condition and incubated at 18–22 °C until germination was detected. Pieces of water agar with teliospores germinating on it were put inversely onto Petri dishes with potato-dextrose-agar (PDA) in order to stimulate the production of secondary sporidia. Nine days later, the Petri dishes were flooded with sterile water and scraped with a sterile spatula, and the suspension was transferred to other Petri dishes with PDA to increase the inoculum. Once the Petri dishes were covered with fungal colonies, the agar was cut into pieces and put inversely on sterile glass Petri dishes, into which distilled water was added and secondary sporidia were daily harvested. Inoculum concentration was adjusted to 10,000 sporidia/mL using a haemocytometer [26].

#### 4.2.2. Karnal Bunt Inoculations and Disease Scoring

Artificial inoculation was performed on two separate occasions for each variety, with specific dates for each variety determined by the availability of sufficient numbers of tillers that were in the boot stage. The inoculations occurred between January and March, and were performed by injecting a sporidial suspension with a hypodermic syringe into the boot just as awns emerged [49]. Appropriate humidity was maintained in the field via intermittent misting with overhead sprinklers during the inoculation period (five times a day, with 20 min of misting each time). At maturity, five inoculated heads were harvested and threshed separately, and the number of infected and uninfected seeds per ear was counted. The data was used to evaluate disease severity, calculated as the percentage of infected grains in each ear. The average of the Karnal bunt infection over five spikes was used in subsequent analysis.

### 4.3. Statistical Analysis

Data analysis was performed using the R statistical software [50]. Analysis of variance was done for estimating the genotypic differences among dates of sowing within a year as well as across the years for the Karnal bunt infection. The Pearson’s correlation coefficient was estimated between different experiments. Variance components were used for estimating the broad-sense heritability H^2^ =σ^2^_g_/(σ^2^_g_ +σ^2^_g*E_/_E_+ σ^2^_e_/_E_), where *σ^2^_g_, σ^2^_g*E_* and σ^2^_e_ represent the genotype, genotype × experiment interaction and residual variance, respectively and E was the number of experiments, respectively.

### 4.4. Genotypic Data and GWAS

Genomic DNA was extracted from young leaves following the previously described CTAB method [51]. The 339 genotypes were genotyped with the DArTseq^®^ technology at the Genetic Analysis Service for Agriculture (SAGA) at CIMMYT, Mexico. This genotyping platform is based on a combination of complexity reduction methods developed for array-based DArT and sequencing of resulting representations on next- generation sequencing platforms, as described in detail in [52]. To avoid spurious marker trait associations, markers with missing data points >20% and minor allele frequency <0.05 were excluded from analysis. Markers that were monomorphic were also removed from the data set and the remaining array comprised 13,098 SNP markers spanning all the chromosomes. Significant marker-trait associations were identified using the mixed linear model approach (MLM) in TASSEL package, which includes a Q matrix for fixed effects and a kinship matrix for random effects. The population structure (Q) and relatedness (K) among genotypes was taken care by the MLM approach to rule out false associations. Marker-trait associations were considered significant if the *p* value of a trait-associated marker is ≤0.001, and were significant in at least two experiments. *R*^2^ was used to evaluate the magnitude of marker trait effects. The marker-trait associations were cross-referenced against all reported QTL in the literature and the GrainGenes database (https://wheat.pw.usda.gov/GG3/). Significant SNPs were used to locate putative candidate genes associated with Karnal bunt resistance, where the SNP sequences were used for BLASTN against the reference genome sequence IWGSC (non-profit organization registered in the United States) RefSeq v1.0 to look for nearby genes associated with plant disease resistance.

## 5. Conclusions

Karnal bunt resistance in common wheat is a complex trait and different genes have been reported controlling the resistance in different accessions. In the present study, eleven lines were identified to be resistant as evident from average Karnal bunt infection scores of <1% after evaluation over four experiments. The lines identified here indicated that variation for Karnal bunt exists and can be used to enrich the Karnal bunt resistance base of breeding germplasm. Markers linked to QTL on chromosomes 2BL and 5BL identified in this study are useful for their utilization in breeding, and further studies on the two most significant QTL could be helpful in the identification of diagnostic markers to be used in marker assisted breeding. The genomic regions spanning the Karnal bunt QTLs identified in our study harbored many genes coding for pathogenesis related proteins such as Leucine-rich repeat receptor-like protein kinase, Kinase family protein, Serine/threonine-protein kinase and Kinase family protein, which have been identified in earlier studies to be conferring resistance. The physical map position of these QTLs linked to these defense related genes would provide opportunities for gene cloning and molecular breeding in wheat.

## Figures and Tables

**Figure 1 ijms-20-03124-f001:**
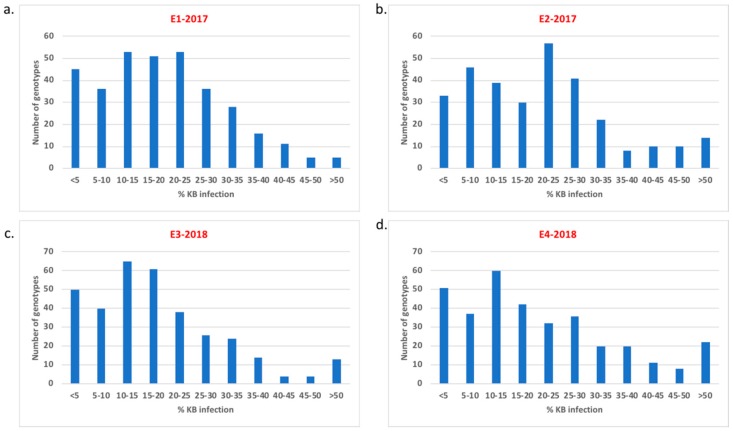
Distribution of Karnal bunt infection over four experiments in the genome-wide association study (GWAS) panel: (**a**) E1-2017 (2017-seeding date 1) (**b**) E2-2017 (2017-seeding date 2) (**c**) E3-2018 (2018-seeding date 1) and (**d**) E4-2018 (2018-seeding date 2).

**Figure 2 ijms-20-03124-f002:**
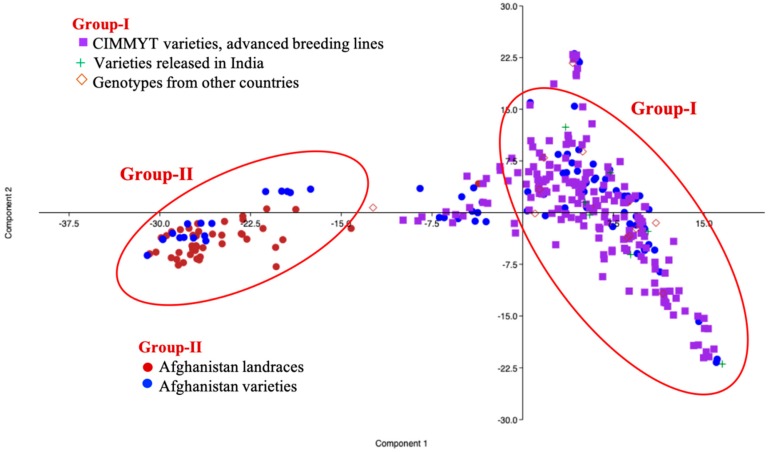
Principal component analysis in the GWAS panel based on genotypic data.

**Figure 3 ijms-20-03124-f003:**
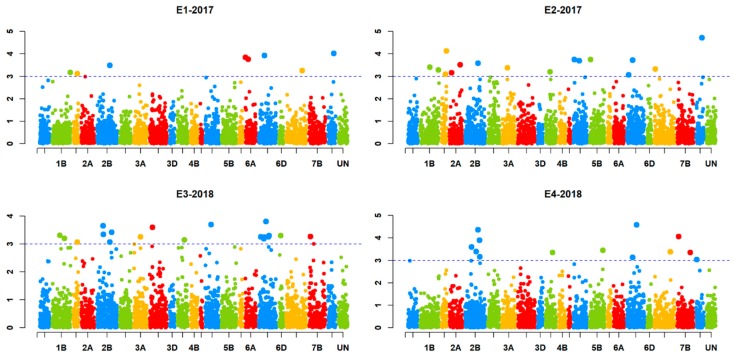
Manhattan plots for significant SNPs associated with Karnal bunt resistance in four experiments.

**Figure 4 ijms-20-03124-f004:**
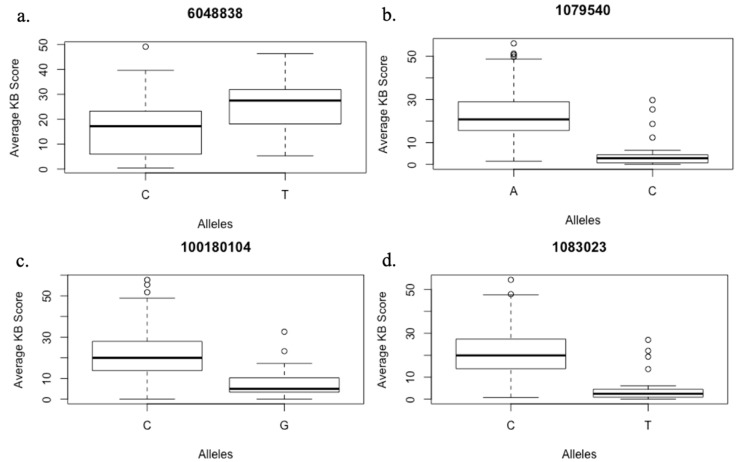
Boxplots showing effect of alleles on the % Karnal bunt infection score based on phenotypic evaluation over four environments (**a**); three environments (**b**) and two environments (**c**,**d**).

**Figure 5 ijms-20-03124-f005:**
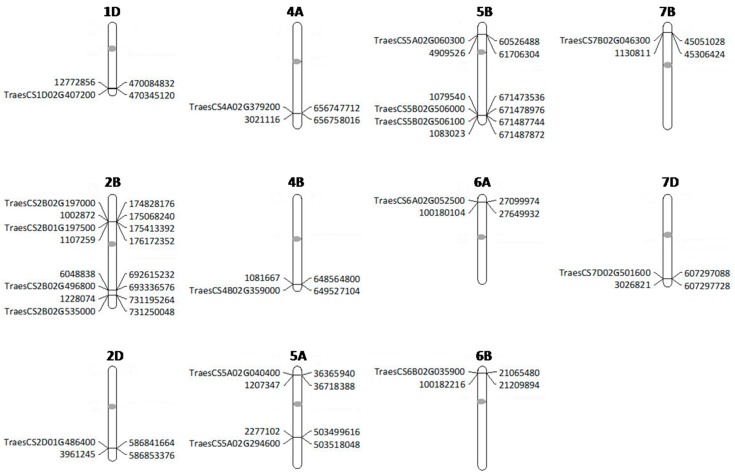
Positions of significant markers and their linked candidate genes projected to the International Wheat Genome Sequencing Consortium Reference Sequence v1.0. Molecular markers and genes are indicated to the left of the chromosomes whereas their physical positions are shown to the right, and the position of centromere is indicated with an oval circle.

**Table 1 ijms-20-03124-t001:** List of reported QTL conferring resistance to Karnal bunt in wheat.

Donor	Chromosome	Linked Marker/ Interval	Reference
Altar 84	3BS, 5AL	RFLP	[30]
HD 29	4BL	*Xgwm* 538	[31]
HD 29	4BL	SNP 52bp fragment of interest (gwm538 snp)	[32]
HD29	Qkb.ksu-5BL.1	*Xgdm*116-Xmc 235	[33]
HD29	Qkb.ksu-6BS.1	*Xwmc*105- *Xgwm* 88	
W485	Qkb.ksu-4BL.1	*Xgwm* 6- *Xwmc* 349	
H567.71	4B	*Xgwm* 6	[34]
ALDAN	Qkb.dwr-5BL.1	*Xwmc* 235 and *Xbarc* 140	[35]
HD29	5B	*Xgdm*116 - *Xwmc*235	[36]
HD29	6B	*Xwmc*105 – *Xgwm* 88	
W485	4B	*Xgwm* 6 – *Xwmc* 349	
WKCBW	*QKb.cim*-2BL	*1086228 - 1092041*	[37]
WKCBW	*QKb.cim*-3DL	*7487658- 2252592*	
Huirivis#1	*QKb.cim-3BS1*	1079551- 100010977	
Mutus	*QKb.cim*-5BS2	2253589 - 1011847	
HD29	3B	*IWB57185*	[38]
WH542	1A	*IWA1644*	
WH542	1D	*IWB2650*	
W485	1B	*IWB59865*	

**Table 2 ijms-20-03124-t002:** List of genotypes from the GWAS panel that showed highly resistant reactions (R) against Karnal bunt infection.

SN	Panel ID	Pedigree	E1-2017	E2-2017	E3-2018	E4-2018	Mean	*QTL
1	498	FAHAD-8-2*2//PTR/PND-T/3/ERIZO-11/YOGUI-3	0	0	0	0	0.0	1, 2, 3
2	516	CMH80A.542/CNO79	0	0	0	0	0.0	1, 3, 4
3	540	ANOAS-5/STIER-13/5/274/320//BGL.3.MUSX/	0	0.48	0	0	0.1	1, 2, 3, 4
4	506	GAUR-2/HARE-3//JLO97/CIVET/5/DIS B5/3/SPHD/	0.65	0	0	0	0.2	1, 2, 3, 4
5	402	ZCL/3/PGFN//CNO67/SON64(ES86-8)/4/SERI/5/UA-2827	0.61	0.53	0	0	0.3	1, 3
6	554	GNU/ASAD//ARDI/3/MANATI-1/4/FAHAD-5	1.78	0	0	0	0.4	1, 2, 3, 4
7	509	PRESTO//2*TESMO-1/MUSX 603/4/ARDI-1/	0	0	0	1.78	0.4	1, 2, 4, 5, 6
8	280	GK.ZOMBOR/ATTILA	1.64	0	0	0.42	0.5	1, 2, 4, 5, 7
9	547	STIER-13/FAHAD-4//MANATI-1/3/POLLMER-1.1	2.05	0	0.4	0	0.6	1, 2, 4, 5
10	558	FD-693/2*FAHAD-4//POLLMER-4/3/POLLMER-2.1	0	2.17	0.96	0	0.8	1, 2, 4
11	6	AFGHAN Wheat Collection #6	0	NA	1.52	1.03	0.9	1, 2, 5, 8, 9, 10, 11, 12

*****QTL composition of the resistant lines: 1. QTL on chromosome 2BS, 2. 5AS, 3. 5BL, 4. 6BS, 5. 2BL, 6. 6AS, 7. 5BS, 8. 5AL, 9. 2BL, 10. 4AL, 11. 4BL, and 12. 7BS.

**Table 3 ijms-20-03124-t003:** Analysis of variance for Karnal bunt infection (%) across four experiments.

Source of Variation	Degree of Freedom	Mean Square	*p* Value	Heritability
Genotype	338	550.71	<0.001	0.80
Experiment	3	437.69	<0.001	
Genotype x Experiment	1352	184.01	<0.001	
Error	1014	61.78		

**Table 4 ijms-20-03124-t004:** Significant and stable SNPs identified for Karnal bunt across at least two experiments (significant at *p* < 0.000).

SN	Marker	Allele	Chromosome	Physical Position	E1- 2017		E2- 2017		E3- 2018		E4- 2018	
					Additve Effect	R^2^	Additive Effect	R^2^	Additive Effect	R^2^	Additive Effect	R^2^
1	12772856	C/T	1DL	470084827	−0.43	0.06	-	-	−2.59	0.06	-	-
2	1002872	A/C	2BS	175068236	-	-	-	-	−5.72	0.06	−6.29	0.04
3	1107259	A/G	2BS	176172344	-	-	-	-	−5.55	0.06	−6.48	0.06
4	1228074	A/G	2BL	731195296	-	-	-	-	−3.47	0.06	−1.25	0.05
5	6048838	T/C	2BL	692615259	−1.90	0.13	−4.64	0.15	−4.27	0.11	−4.96	0.20
6	3961245	T/C	2DL	586853396	-	-	−1.75	0.06	−1.54	0.05	-	-
7	3021116	A/C	4AL	656758037	-	-	-	-	0.75	0.06	1.80	0.07
8	1081667	A/G	4BL	648564812	−4.13	0.05	−2.96	0.05	-	-	-	-
9	1207347	G/A	5AS	36718388	−3.55	0.05	−4.05	0.07	-	-	-	-
10	2277102	T/C	5AL	503499615	-	-	0.46	0.06	−0.57	0.06	-	-
11	1079540	C/A	5BL	671473536	1.25	0.05	1.69	0.06	2.75	0.06	-	-
12	1083023	C/T	5BL	671487862	NaN	0.06	-	-	NaN	0.06	NaN	0.08
13	4909526	C/G	5BS	61706304	5.83	0.07	-	-	-	-	7.76	0.08
14	100180104	G/C	6AS	27649931	4.69	0.07	3.32	0.06	-	-	-	-
15	4394191	G/A	6BL	500595153	-	-	7.69	0.10	0.91	0.06	-	-
16	100182216	T/G	6BS	21209894	-	-	-	-	4.87	0.07	7.08	0.09
17	1130811	G/C	7BS	45306426	-	-	-	-	−1.58	0.05	−1.78	0.06
18	3026821	C/G	7DL	607297738	3.10	0.06	2.96	0.08	-	-	-	-

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
