# Peer review of "Genome Wide Association Study of Karnal Bunt Resistance in a Wheat Germplasm Collection from Afghanistan"

_ijms, 2019, doi:10.3390/ijms20133124_

Reviewer 1 Report

MS is very interesting and valuable. I have a few minor concerns those can be addressed in the revised version.

Abstract

“with respect to the strict quarantine regulations on import and export of affected seeds” – this will be misleading, two issues (1) the article is not addressing issues of quarantine regulations, (2) miss-identify the challenge, the real challenge is to eradicate or limit the disease.

 “only a few genotypes have been found to carry resistance” – partial resistance or complete resistance?

 I recommend not to use abbreviation KB for the Karnal bunt, instead write it fully, Authors can use KBR when talking about resistance. KB is confusing for genomics person.

 Authors can use “most significant loci” for the QTL, it is preferential if want to keep QTL, no issue.

 Phenotypic variation governed by the loci is a big concern for breeder community but the estimated PV is always depending on the genetic background, QTLs sowing minor PV can show over 50% PV when segregated in near-isogenic background. My point is that here in this MS no need to highlight PV too much.

 Introduction

Better to use consistently Neovossia indica (Mitra) in Abstract also.

More information about GWAS/QTL mapping and genotyping approaches can be added. The author can find comparative information here doi: 10.1111/pbi.12249.

 Table 1 – need better formatting/ presentation

 Table 2 is not very informative, I suggest removing it and provide the values in the text (which are already there).

 The basis behind the level of significance (p-3) chosen need to be discussed. What will be the scenario if authors select Bonferroni correction?

Conclusion section can be expanded further to incorporate more about the candidate genes identified in significant and stable loci.

Author Response

Reviewer 1:

Comments and Suggestions for Authors

MS is very interesting and valuable. I have a few minor concerns those can be addressed in the revised version.

1.       Abstract

a)       “with respect to the strict quarantine regulations on import and export of affected seeds” – this will be misleading, two issues (1) the article is not addressing issues of quarantine regulations, (2) miss-identify the challenge, the real challenge is to eradicate or limit the disease.

Response:

Yes the real challenge is to eradicate the disease, but this disease gained importance because of strict quarantine regulations for the movement of infected seed. This disease is not causing much grain yield losses it affects the grain quality due to its fishy odour in infected seeds. We have tried to give focus to both resistance and quarantine regulations in the manuscript.

 b)      “only a few genotypes have been found to carry resistance” – partial resistance or complete resistance?

Response:

The identified genotypes had only partial resistance due to the polygenic nature of trait and this point has been addressed in the manuscript.

 c)       I recommend not to use abbreviation KB for the Karnal bunt, instead write it fully, Authors can use KBR when talking about resistance. KB is confusing for genomics person.

 Response:

The abbreviation KB have been removed and Karnal bunt has been consistently used in the manuscript.

d)      Authors can use “most significant loci” for the QTL, it is preferential if want to keep QTL, no issue.

Response:

We want to keep QTL for the significant loci in the manuscript.

 e)       Phenotypic variation governed by the loci is a big concern for breeder community but the estimated PV is always depending on the genetic background, QTLs sowing minor PV can show over 50% PV when segregated in near-isogenic background. My point is that here in this MS no need to highlight PV too much.

Response:

Yes it is true that background of a genotype matters a lot for the expression of transferred gene/QTL. We have used PV only for the two main QTLs identified in the study to reflect the variation explained by these QTLs only.

 2.       Introduction

Better to use consistently Neovossia indica (Mitra) in Abstract also.

Response:

Thanks, Neovossia indica has been used in the revised manuscript.

3.       More information about GWAS/QTL mapping and genotyping approaches can be added. The author can find comparative information here doi: 10.1111/pbi.12249.

 Response:

Additional information about the advantages of GWAS over QTL mapping has been updated in the manuscript as suggested.

 4.       Table 1 – need better formatting/ presentation.

Response:

In Table 1, only the significant QTLs identified in the previous studies has been retained and the minor regions identified were removed, making it more precise and presentable.

5.       Table 2 is not very informative, I suggest removing it and provide the values in the text (which are already there).

 Response:

Table 2 has been deleted from the manuscript as suggested.

 6.       The basis behind the level of significance (p-3) chosen need to be discussed. What will be the scenario if authors select Bonferroni correction Response:

 Response:

The critical p value we adopted (0.001) was based on a balance between false positive and false negative and is being used by many researchers in the crop genetic area. In the early stage of GWAS, some people use p values of 0.05 or 0.01, which is too high and could lead to numerous false positive results; later when people realized this, a very stringent strategy called Bonferroni correction was adopted, in which the p value was determined by 0.05/No. of markers. In our study, the p value would be 0.05/13098=3.8 x 10-6 if the Bonferroni correction were used, which is too low and would result in the absence of any significant QTL (see Fig. 3). As demonstrated in the manuscript, we’ve identified several QTL that have been reported previously, which implies that our analysis is robust, and also our critical p value is appropriate. Inheritances of many traits in wheat, as well as in other crops, are of quantitative, meaning that many loci with minor to moderate phenotypic effects are involved, if a very low p value were used as the critical threshold, then very few or no loci would be detected. Therefore, nowadays most researchers use p=0.001 or 0.0001, instead of p values from Bonferroni correction.

 7.       Conclusion section can be expanded further to incorporate more about the candidate genes identified in significant and stable loci.

Response:

The information about candidate genes identified in the region spanning the QTLs is included and future prospects about cloning and use in molecular breeding has been updated.

Reviewer 2 Report

This manuscript presents very useful information regarding potential QTLs, gene candidates, and germplasm to aid in breeding resistance to Karnal bunt infection into wheat. The manuscript is generally very well written, but there are a number of small grammatical errors throughout such as the following:

L16 …the fungus…

L19 …a few genotypes…

L27 Germplasm with…

A quick edit will make the manuscript much easier to read, though, as I say, it is already quite good and easy to follow.

In Figure 2, the legend and the PCA symbols for "varieties released in India" don’t quite match. It isn’t a huge issue, but if it is easy to fix, it would make it easier to interpret the PCA.

Given that the experiment was conducted in one location in two years, I would like to know how the heading dates differed as a result of the two different planting dates. For these to be considered statistically as different site-years, I would expect a significantly different boot date when the germplasm was susceptible to infection. Otherwise, they only represent additional replications in the two different site-years. One way to present this would be the actual inoculation dates for the two planting dates as a plot showing range of first and second inoculations for each of the planting dates.

It isn’t essential for the manuscript to be useful, but it would add some additional justification for the two planting dates consideration as separate site-years.

Author Response

Reviewer 2:

Comments and Suggestions for Authors :

 1.       This manuscript presents very useful information regarding potential QTLs, gene candidates, and germplasm to aid in breeding resistance to Karnal bunt infection into wheat. The manuscript is generally very well written, but there are a number of small grammatical errors throughout such as the following:

L16 …the fungus…

L19 …a few genotypes…

L27 Germplasm with…

A quick edit will make the manuscript much easier to read, though, as I say, it is already quite good and easy to follow.

Response:

The manuscript has been revised and grammatical errors have been removed.

 2.       In Figure 2, the legend and the PCA symbols for "varieties released in India" don’t quite match. It isn’t a huge issue, but if it is easy to fix, it would make it easier to interpret the PCA.

 Response:

The figure has been revised and the symbols have been fixed in the figure.

 3.       Given that the experiment was conducted in one location in two years, I would like to know how the heading dates differed as a result of the two different planting dates. For these to be considered statistically as different site-years, I would expect a significantly different boot date when the germplasm was susceptible to infection. Otherwise, they only represent additional replications in the two different site-years. One way to present this would be the actual inoculation dates for the two planting dates as a plot showing range of first and second inoculations for each of the planting dates. It isn’t essential for the manuscript to be useful, but it would add some additional justification for the two planting dates consideration as separate site-years

 Response:

In each year two experiments were conducted, the first experiment comprising of sowing on mid-November while the second experiment was planted two weeks later. The environmental conditions in which the experiments were conducted will be different due to the difference of two weeks. Hence cannot be taken as replication and considered them as independent experiments.

The graph representing inoculation days for each entry experiment wise within each year has been included as supplementary figure S1.